# Determinants of diabetic nephropathy among diabetic patients in Ethiopia: Systematic review and meta-analysis

**Abere Woretaw Azagew** [ID]*, **Zerko Wako Beko, Chilot Kassa Mekonnen**

Department of Medical Nursing, School of Nursing, College of Medicine and Health Sciences, University of Gondar, Gondar, Ethiopia

* wabere@ymail.com

## Abstract

### Introduction

Diabetic nephropathy (DN) is a long-term kidney disease among diabetic patients. It is the leading cause of end-stage renal failure. In Ethiopia, DN affects the majority of diabetic populations, but there were inconsistent findings about the determinant factors across the studies.

### Methods

We have accessed studies using PubMed, Embase, EBSCO, Web of Science, OVID, and search engines including Google and Google Scholar published up to June 2023. The study populations were diabetic patients with nephropathy. The quality of each included article was assessed using the Newcastle-Ottawa quality assessment scale. The odds ratios of risk factors were pooled using a random-effect meta-analysis model. Heterogeneity was assessed using the Cochrane Q statistics and I-Square ($I^2$). The publication bias was detected using the funnel plot and/or Egger's test ($p < 0.05$). Trim and fill analysis was carried out to treat the publication bias. The protocol has been registered with the reference number CRD42023434547.

### Results

A total of sixteen articles were used for this reviewed study. Of which, eleven articles were used for advanced age, ten articles for duration of diabetic illness, ten articles for poor glycemic control, and eleven articles for having co-morbid hypertension. Diabetic patients with advanced age (AOR = 1.11, 95% CI: 1.03–120, $I^2$ = 0.0%, p = 0.488), longer duration of diabetic illness (AOR = 1.23, 95% CI = 1.05–1.45, $I^2$ = 0.0%, p = 0.567), poor glycemic control (AOR = 2.57, 95% CI: 1.07–6.14; $I^2$ = 0.0%, p = 0.996), and having co-morbid hypertension (AOR = 4.03, 95% CI: 2.00–8.12, $I^2$ = 0.0%, p = 0.964) were found to be factors associated with DN.

### Conclusions

The findings of the study revealed that diabetic patients with advanced age, longer duration of diabetic illness, poor glycemic control status, and co-morbid hypertension were the

---

**Data Availability Statement:** All relevant data are within the manuscript and its Supporting Information files.

**Funding:** The author(s) received no specific funding for this work.

---

**Competing interests:** The authors have declared that no competing interests exist.

**Abbreviations:** AOR, Adjusted Odds Ratio; CI, Confidence Interval; Co-HTN, co-morbid Hypertension; DN, Diabetic Nephropathy; DKD, diabetic Kidney disease; GFR, Glomerular filtration rate; OR, Odds Ratio; PRISMA, Preferred Reporting Item of Systematic Review and Meta-Analysis.

determinant factors of DN. Therefore, treatment of co-morbid hypertension and high blood glucose and regular screening of renal function should be implemented to detect, treat, and reduce the progression of DN. Furthermore, healthcare workers should give due attention to diabetes with advanced age and a longer duration of diabetes illness to prevent the occurrence of DN.

## Introduction

Diabetes mellitus is a major public health problem worldwide. The global prevalence was 10.5% in 2021 and has been projected to be 12.2% in 2045 [1]. In Ethiopia, it ranged from 2.0% to 6.5% [2]. The mortality and morbidity of diabetes mellitus are associated with chronic complications, such as vascular and non-vascular. Diabetic nephropathy (DN) is one of the major micovascular complications that leads to end-stage renal disease [3,4]. It is the leading public health challenge globally. Diabetic nephropathy is also known as diabetic kidney disease (DKD). Clinically, DN is characterized by an increase in urine albumin excretion (microalbuminuria) and/or a decreased glomerular filtration rate (GFR) [5–7]. It is the only complication for diabetic populations compared with non-diabetics [8]. The causes of DN are multifactoral, but they are associated with being overweight, diet, exercise, smoking, co-morbidity, type of diabetes, and non-adherence to diabetic medications [5,7,9–14].

Diabetic nephropathy is increasing significantly in low- and middle-income countries, while the remaining is under-recognized as a global illness burden [6,15]. The median time for diabetic patients to develop DN was 94.9 months [10]. Its prevalence was 21.8% in China [12] and 10.8% in Saudi Arabia [14]. Overall, DN occurs in 20–40% of all types of diabetics [16].

Diabetes and its complications affect individuals, families, healthcare systems, and national economies [17]. The World Health Organization-led initiative is targeted at reducing diabetic-associated kidney disease by providing comprehensive care [18]. Preventive strategies improve glycemic control by adhering to medication, food, and exercise [5]. Early detection and screening for microalbuminuria should be done regularly [6,7]. In advanced cases, patients require renal replacement therapy (RRT), such as hemodialysis, peritoneal dialysis, or kidney transplantation [19].

Though various studies have been undertaken on DN, the knowledge of the determinant factors of DN has been inconsistent across studies among diabetic patients. Therefore, this review study aimed to investigate the determinants of DN among diabetic patients in Ethiopia.

## Methods

### Study protocol registration and reporting

The review protocol has been registered with reference number CRD42023434547. The result of the review was presented based on the standard of preferred reporting items for systematic review and meta-analysis (PRISMA-2020) checklist [20] (**S1 File**).

### Searching strategies

The procedure for this study was designed following the PRISMA guidelines [20]. We search on electronic databases such as PubMed, Embase, EBSCO, Web of Science, OVID, and search engines (Google and Google Scholars) for grey literature. Endnote (version 7) reference management software was used to export, download, organize, review, de-duplicate, and cite the

articles. The search strategies were carried out using controlled vocabulary (MeSH) terms. The synonym for diabetic nephropathy was identified. Then, the search string was established using the databases. Articles were searched by title (ti), abstract (ab), and/or full-text (ft). Boolean logic operators "AND" and "OR" were used to combine the search terms. The search string was expressed as "diabetic nephropath*" OR "diabetic kidney disease" OR DKD OR "chronic kidney disease" OR CKD OR "diabetic glomerulosclerosis" OR "Kimmelstiel-Wilson disease" OR "Kimmelstiel-Wilson Syndrome" OR "nodular glomerulosclerosis" AND "diabetic mellitus" OR "Type 1 diabetic mellitus" OR "T1DM" OR "Type 2 diabetic mellitus" OR T2DM, OR Hyperglycemia, OR "elevated blood glucose" AND Ethiopia. This search included articles published up to June 2023.

The search limiters, such as study design, country of the study, and language of publication, were used. Furthermore, to include additional relevant articles, citation tracking and personal contact with experts were carried out. Two reviewers (ZWB and CKM) screened articles by title (ti), abstract (ab), and full-text (ft). The disagreements between the reviewers were resolved by discussion.

## Study population

The populations for this systematic review and meta-analysis were diabetic patients with nephropathy.

## Inclusion and exclusion criteria

In this systematic review and meta-analysis, articles that fulfill the eligibility criteria: (1) articles with observational studies such as cross-sectional, cohort, and case-control; (2) articles that report the determinants, risk factors, or associated factors of diabetic nephropathy; (3) studies conducted in Ethiopia; (4) both published and/or unpublished articles; (5) articles published or conducted up-to June 2023; (6) articles written in English; and (7) articles conducted in community or health facility settings were included in the study. Articles with trials, without full text, conference papers, and systematic reviews and meta-analyses were excluded from the study.

## Quality appraisal

The quality of the study was evaluated using the Newcastle Ottawa Quality Assessment Scale (NOS), adapted from the cross-sectional [21], cohort [22], and case-control [23] studies. Two reviewers (ZWB and CKM) independently reviewed the quality of the included studies. The inconveniences between the reviewers were resolved by discussion. A score of $\geq 7$ was considered a high-quality score [24].

## Outcome measurements

Diabetic nephropathy is defined as a derangement in renal function with an estimated GFR $<60$ ml/min/1.73 m$^3$, and kidney damage, usually by estimation of albumin uria $>30$ mg/dl [25].

## Data extraction

After the quality assessment of the studies, the data were extracted using the Microsoft Excel speed sheet. The extracted data item includes authors, publication year, study design, region of the study, data collection method, funding source, and other effect measures (**S2 File**). Two

reviewers (ZWB and CKM) independently extracted the data. The disagreements between the reviewers were resolved through discussion and third-reviewer involvement.

## Data analysis

The extracted data were entered into a Microsoft spreadsheet and exported to Stata version 14 for analysis. The heterogeneity of the studies was assessed using the I-squared ($I^2$) and Cochrane Q statistics with a p-value > 0.05. The $I^2$ test statistics result of 0.0%, 25%, 50%, and >75% were considered as no, low, moderate, and considerable heterogeneity [26], respectively. A random-effect meta-analysis model [27] was used to estimate the effect of determinant factors on diabetic nephropathy. The forest plot was used to visualize the pooled effect size with the respective odds ratio and 95% confidence interval. The publication bias was detected by the visual inspection of the graphic asymmetry test of the funnel plot [28] and/or Egger's test (P<0.05) [29]. Trim and fill analysis was computed to treat the publication bias [30].

## Results

### Study selection and characteristics

The search strategy retrieved 1031 both published and/or unpublished original articles. After the removal of duplicate articles, 113 articles remained. Following further screening, about 19 articles were assessed for eligibility, of which three were excluded because of reporting without the outcome of interest (the DN is not clearly defined) and poor quality (the NOS score <5) (**Fig 1**). Finally, sixteen articles were used to extract the determinant factors of DN. Of these, eleven articles were retrieved for age of diabetic patients [5,9,31–39] (**Table 1**), ten for duration of diabetic illness [5,9,15,31,32,36,38–41] (**Table 2**), ten for poor glycemic control [5,9,32–34,36,39,41–43] (**Table 3**), eight for elevated systolic blood pressure [9,15,33,34,36,38,41,44] (**Table 4**), and twelve for co-morbid hypertension [5,15,31,33,36,38–44] (**Table 5**). From the included studies, seven were in the Amhara region [15,31,34,38,39,41,44], four in the Oromia region [33,36,37], two in Addis Ababa [40,42], two in the Tgray region [5,9], and two in the SNNP region [32,43] of Ethiopia. Regarding the study design, as shown in each table, most of the studies were cross-sectional studies. The majority of the studies used both patient interviews and record reviews for data collection. Most of the included studies were not funded or their funding statuses were unreported. The average quality appraisal score of the included studies was ≥7.

### Determinant factors of diabetic nephropathy

**Age of diabetic patients.** Diabetic patients with older age are 1.11 times more likely to develop DN compared to those diabetic patients with younger age (AOR = 1.11, 95% CI: 1.03–120, $I^2$ = 0.0%, P = 0.488) (**Fig 2**).

**Heterogeneity test and publication bias.** The overall heterogeneity of the included study showed that $I^2$ = 0.0% and the Cochrane Q statistic p-value = 0.488 using a random effect model, indicating there is no variation across studies. Regarding the publication bias, Egger's test p-value is 0.005, and the funnel plot test result indicates asymmetrical (**Fig 3**) meaning that there is a publication bias. In order to treat the bias, a trim and fill analysis was carried out. Two iteration cycles were performed, and six articles were added, which made a total of 17 articles with 16 degrees of freedom, a p-value of 0.518, and a moment-based between study variance of 0.0 (**Fig 4**).

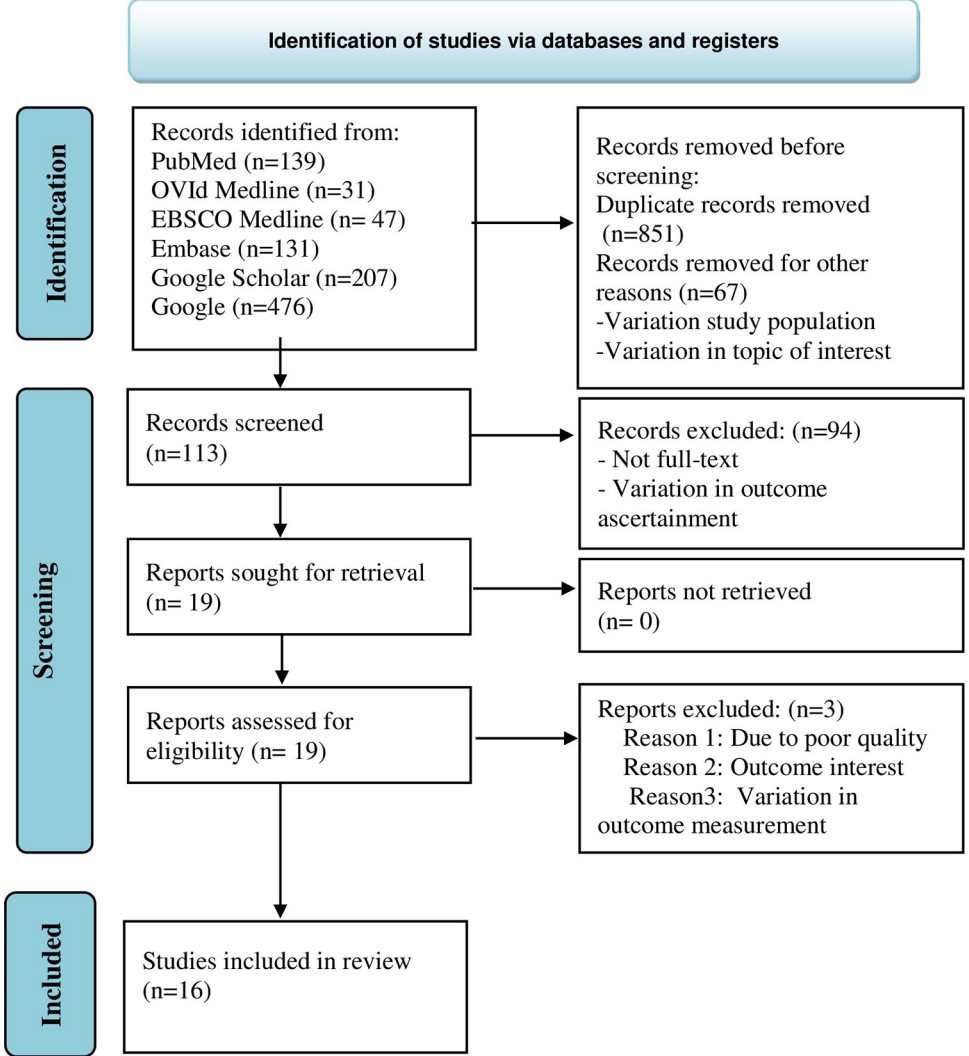

**Fig 1. PRISMA flow diagram for the flow of information through the phases of the review.**

**Duration of diabetic illness.** Diabetic patients with a longer duration of diabetic illness are 1.23 times more likely to develop DN compared with diabetic patients with a shorter duration of diabetic illness (AOR = 1.23, 95% CI = 1.05–1.45, $I^2$ = 0.0%, p = 0.567) (**Fig 5**).

**Heterogeneity test and publication bias.** The overall heterogeneity test of the included studies showed that $I^2$ is nil and the Cochrane Q statistic p-value is 0.567 using a random effect model, indicating the included studies are homogenous. Concerning the publication bias, Egger's test p-value is 0.004, and the graphic funnel plot test showed symmetrical (**Fig 6**), indicating there is publication bias across the studies. To treat the bias, trim and fill analysis was carried out, and two iteration cycles were performed, and six articles were added, making a total of 16 articles with 15 degrees of freedom, a p-value of 0.741, and a moment-based between study variance of 0.0 (**Fig 7**).

**Poor glycemic control.** Diabetic patients who have poor glycemic control status are 2.57 times more likely to develop DN compared to those diabetic patients with good glycemic control status (AOR = 2.57, 95% CI: 1.07–6.14; $I^2$ = 0.0%, p = 0.996) (**Fig 8**).

**Table 1. Study characteristics for the age of diabetic patients in Ethiopia.**

| Authors | year | study design | Region | Odds ratio | 95% CI | quality score |
|---|---|---|---|---|---|---|
| Damtie S, et al. [31] | 2018 | Cross-sectional | Amhara | 5.239 | 2.255–12.175 | 9 |
| Fiseha T, et al. [32] | 2014 | Cross-sectional | SNNP | 5.3 | 1.81–15.56 | 8 |
| Dinku B, et al. [33] | 2022 | Cross-sectional | Oromia | 2.17 | 1.09–4.31 | 8.5 |
| Hintsa S, et al. [9] | 2017 | Case-control | Tigray | 1.037 | 1.01–1.064 | 7 |
| Mulu GB, et al. [34] | 2023 | Cross-sectional | Amhara | 4.1 | 2.2–7.7 | 7.5 |
| Zemichael TM, et al. [5] | 2020 | Case-control | Tigray | 1.19 | 1.16–1.23 | 8 |
| Abdulkadir M, et al. [35] | 2022 | Cross-sectional | Addis Ababa | 5.8 | 1.5–21 | 8 |
| Adem M, et al. [36] | 2017 | Cross-sectional | Oromia | 3.02 | 1.55–5.9 | 7.5 |
| Goro KK, et al. [37] | 2019 | Cross-sectional | Oromia | 2.01 | 1.1–5 | 8.5 |
| Tesfe D, et al. [38] | 2022 | Cross-sectional | Amhara | 5.74 | 3.05–10 | 9 |
| Fiseha T, etal. [39] | 2020 | Cross-sectional | Amhara | 2.48 | 1.13–5.43 | 8 |

Notes:- **CI-Confidence interval; St PMMC**- Siant Paulos Millennium Medical College; **SNNP**-Southern Nations, Nationalities and Peoples.

**Heterogeneity test and publication bias.** The heterogeneity test of the studies indicates that $I^2 = 0.0\%$ and the Cochrane Q statistic p-value is 0.996 using a random effect model, meaning that there is no variation across the included studies. The publication bias test result of Egger's test p-value is 0.07, and the funnel plot test looks symmetrical (**Fig 9**), indicating there is no publication bias.

**Elevated systolic hypertension.** Diabetic patients with elevated systolic hypertension is not at risk for developing DN (AOR = 1.82, 95%CI: 0.83–3.98; $I^2 = 0.0\%$, p = 0.848) (**Fig 10**).

**Co-morbid hypertension.** Diabetic patients with co-morbid hypertension are 4.03 times more likely to develop DN compared to diabetic patients without co-morbid hypertension (AOR = 4.03, 95%CI: 2.00–8.12, $I^2 = 0.0\%$, p = 0.964) (**Fig 11**).

**Heterogeneity test and publication bias.** As shown in the forest plot in **Fig 11** above, the $I^2 = 0.0\%$ and the Cochrane Q statistic p-value is 0.964 using a random effect model, indicating there is no publication bias. Regarding the publication bias, the graphic asymmetry test of the funnel plot showed a symmetrical distribution (**Fig 12**), and the Eggers test p-value is 0.817, indicating there is no publication bias.

**Table 2. Study characteristics for the duration of diabetic illness among diabetic patients in Ethiopia.**

| Authors | year | study design | Region | Odds ratio | 95% CI | quality score |
|---|---|---|---|---|---|---|
| Alemu H, et al. [15] | 2020 | Cross-sectional | Amhara | 3.2 | 2.0–7.0 | 9 |
| Bekele MM, et al. [40] | 2016 | Cross-sectional | Addis Ababa | 1.45 | 0.73–2.88 | 8 |
| Damtie S, et al. [31] | 2018 | Cross-sectional | Amhara | 3.38 | 1.393–8.198 | 9 |
| Fiseha T, et al. [32] | 2014 | Cross-sectional | SNNP | 4.08 | 1.7–9.77 | 8 |
| Hintsa S, et al. [9] | 2017 | Case-control | Tigray | 1.13 | 1.077–1.18 | 7 |
| Taderegew MM, et al. [41] | 2020 | Cross-sectional | Amhara | 3.38 | 1.45–7.82 | 8 |
| Zemichael TM, et al. [5] | 2020 | Case-control | Tigray | 1.83 | 1.62–2.06 | 8 |
| Adem M, et al. [36] | 2017 | Cross-sectional | Oromia | 4.44 | 2.3–8.57 | 7.5 |
| Tesfe D, et al. [38] | 2022 | Cross-sectional | Amhara | 3.37 | 1.86–6.1 | 9 |
| Fiseha T, et al. [39] | 2020 | Cross-sectional | Amhara | 5.16 | 1.13–12.51 | 8 |

Notes:- **CI**- Confidence interval; **SNNP**-Southern Nations, Nationalities and Peoples.

**Table 3. Study characteristics for patients with poor glycemic status among diabetic patients in Ethiopia.**

| Authors | year | study design | Region | Odds ratio | 95% CI | quality score |
|---|---|---|---|---|---|---|
| Aberra T, et al. [42] | 2022 | Cross-sectional | Addis Ababa | 1.23 | 0.667–2.276 | 9 |
| Fiseha T, et al. [32] | 2014 | Cross-sectional | SNNP | 4.65 | 1.69–12.76 | 8 |
| Dinku H, et al. [33] | 2022 | Cross-sectional | Oromia | 1.37 | 0.75–2.49 | 7.5 |
| Hintsa S, et al. [9] | 2017 | Case-control | Tigray | 2.71 | 1.49–4.95 | 7 |
| Mulu GB, et al. [34] | 2023 | Cross-sectional | Amhara | 2.5 | 1.5–4.1 | 7.5 |
| Taderegew MM, et al. [41] | 2020 | Cross-sectional | Amhara | 2.82 | 1.13–7.05 | 8 |
| Zemichael TM, et al. [5] | 2020 | Case-control | Tigray | 3.27 | 1.31–8.31 | 8 |
| Adem M, et al. [36] | 2017 | Cross-sectional | Oromia | 7.24 | 2.16–24.22 | 7.5 |
| Nibret E, et al. [43] | 2020 | Cohort | SNNPR | 9.09 | 4.0–20.0 | 8.5 |
| Fiseha T, et al. [39] | 2020 | Cross-sectional | Amhara | 2.58 | 1.19–5.61 | 8 |

**Notes: CI**- Confidence interval; **SNNP**-Southern Nations, Nationalities and Peoples.

## Discussion

This systematic review and meta-analysis study identifies the important risk factors for DN. The findings of this study revealed that advanced age (≥55 years) is a risk factor for DN. The findings of the study is supported by the study conducted in Italy [45]. This is due to the fact that age plays a vital role in the deterioration of renal-vasculature functionality [46]. This problem worsens in the diabetic patient population [47]. This is due to advanced age, which results in stiffness of the blood vessels, eventually leading to renal vascular complications [48]. Furthermore, aging can predispose to various co-morbidities linked to several factors, such as oxidative stress, apoptosis, myocardial deterioration, and degeneration [49].

The duration of diabetic illness is the determinant factor for the development of DN. Patients with a longer duration of diabetic illness (≥10 years) are at risk of developing DN. This is consistent with the study conducted in the Netherlands [50]. This is due to the fact that uncontrolled blood glucose along with a longer duration of diabetic illness leads to renal vasculature damage, which impairs renal function [51].

The findings also depict that glycemic control status is the determinant factor of DN. Patients with poor glycemic control are at risk of developing DN. This is supported by the study in Iran [52]. Low socioeconomic status in medical care is responsible for poor glycemic control among diabetic patients [53]. Over time, poorly controlled diabetes can cause damage

**Table 4. Study characteristics for patients with elevated systolic blood pressure among diabetic patients in Ethiopia.**

| Authors | year | study design | Region | Odds ratio | 95% CI | quality score |
|---|---|---|---|---|---|---|
| Dinku B, et al. [33] | 2022 | Cross-sectional | Oromia | 3.2 | 1.36–7.51 | 8.5 |
| Hintsa S, et al. [9] | 2017 | Case-control | Tigray | 2.78 | 1.69–4.58 | 7 |
| Mulu GB, et al. [34] | 2023 | Cross-sectional | Amhara | 1.9 | 1.0–3.4 | 7.5 |
| Taderegew MM, et al. [41] | 2020 | Cross-sectional | Amhara | 0.78 | 0.49–1.07 | 8 |
| Alemu H, et al. [15] | 2020 | Cross-sectional | Amhara | 6.0 | 4.0–22.0 | 8.5 |
| Adem M, et al. [36] | 2017 | Cross-sectional | Oromia | 2.19 | 1.12–4.28 | 7.5 |
| Tesfe D, et al. [38] | 2022 | Cross-sectional | Amhara | 6.33 | 3.34–11.99 | 9 |
| Adem M, et al. [44] | 2021 | Cross-sectional | Amhara | 0.997 | 0.594–1.72 | 8 |

**Notes:**—**CI**-Confidence interval.

**Table 5. Study characteristics for patients' co-morbid hypertension among diabetic patients in Ethiopia.**

| Authors | year | study design | Region | Odds ratio | 95% CI | quality score |
|---|---|---|---|---|---|---|
| Aberra T, et al. [42] | 2022 | Cross-sectional | Addis Ababa | 1.37 | 0.865–2.169 | 8.5 |
| Adem M, et al. [44] | 2021 | Cross-sectional | Amhara | 2.279 | 1.025–5.067 | 8 |
| Alemu H, et al. [15] | 2020 | Cross-sectional | Amhara | 8.2 | 2.0–23 | 8.5 |
| Bekele MM, et al. [40] | 2016 | Cross-sectional | Addis Ababa | 1.16 | 0.81–3.26 | 7 |
| Damtie S, et al. [31] | 2018 | Cross-sectional | Amhara | 4.51 | 2.266–8.977 | 9 |
| Dinku B, et al. [33] | 2022 | Cross-sectional | Oromia | 4.89 | 1.93–12.4 | 8.5 |
| Taderegew MM, et al. [41] | 2020 | Cross-sectional | Amhara | 3.12 | 1.51–6.45 | 8 |
| Zemichael TM, et al. [5] | 2020 | Case-control | Tigray | 6.44 | 4.46–9.28 | 8 |
| Adem M, et al. [36] | 2017 | Cross-sectional | Oromia | 5.62 | 2.81–11.23 | 7.5 |
| Tesfe D, et al. [38] | 2022 | Cross-sectional | Amhara | 4.85 | 2.07–11.3 | 9 |
| Nibret E, et al. [43] | 2020 | Cohort | SNNPR | 12.5 | 8.33–25.0 | 8.5 |
| Fiseha T, et al. [39] | 2020 | Cross-sectional | Amhara | 3.37 | 1.45–7.86 | 8 |

**Notes:-CI**-Confidence interval; **SNNP**-Southern Nations, Nationalities and Peoples.

to blood vessel clusters in the kidneys that filter waste. High blood glucose affects the microvasculature of the kidneys, which results in nephrosclerosis [54].

Furthermore, co-morbid hypertension is found to be a risk factor for DN. The result of this study is supported by the studies conducted in Sub-Saharan Africa [55] and Australia [56]. This is because in Sub-Saharan studies, the similarity is due to poor lifestyle intervention and low socioeconomic status to prevent the occurrence of co-morbid hypertension [57]. The relationship between co-morbid hypertension and DN is not well investigated,

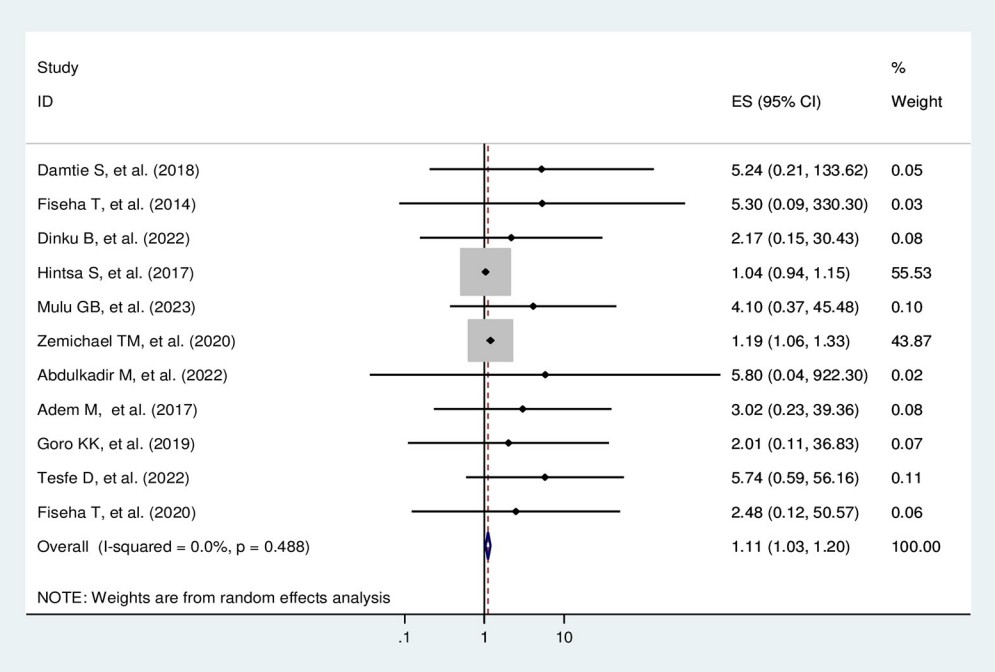

**Fig 2. The pooled effect of age on diabetic nephropathy.**

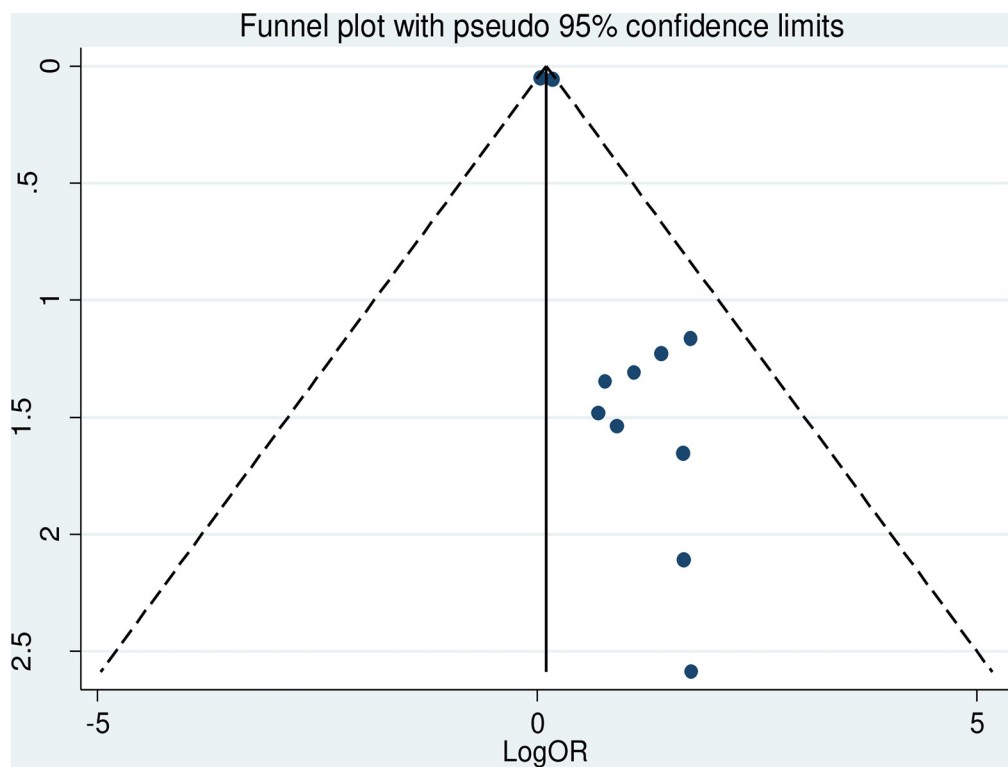

**Fig 3. Funnel plot shows the distribution of included studies.**

but it is associated with renal artery sclerosis, and metabolic abnormalities derived from diabetes eventually lead to renal function deterioration [58]. Elevated hypertension is directly transmitted to the glomeruli. This leads to glomerular hypertension and the activation of mediators that induce inflammation, fibrosis, and further injury of the glomeruli [59]. High blood pressure causes further kidney damage by increasing the pressure in the delicate system of the kidney [60].

As a strong point, the authors used some of the important major databases to search for original research articles. In addition, the authors tried to incorporate different factors to estimate their effect on DN. On the contrary, the findings of this study have important limitations. 1) The study only contains observational study designs; 2) restricted only to the English language; 3) focused on the determinant factors of DN but not its prevalence; 4) the authors include both type 1 and/or type 2 diabetic patients; and 5) it incorporates specific factors. Therefore, the authors recommended further research that includes other than observational study designs; conduct on type 1 and type 2 diabetes patients separately; and other predictors for DN that need to be studied.

## Implications of the study

Nowadays, diabetic kidney disease is the main public health concern. As a result, the findings of this study provide input for policymakers to design effective diabetic care and complication prevention strategies. It also helps healthcare workers implement evidence-based interventions for diabetic patients with advanced age, living with a longer duration of diabetic illness, having poor glycemic control status, and having co-morbid hypertension to tackle the occurrence, burden, and severity of DN.

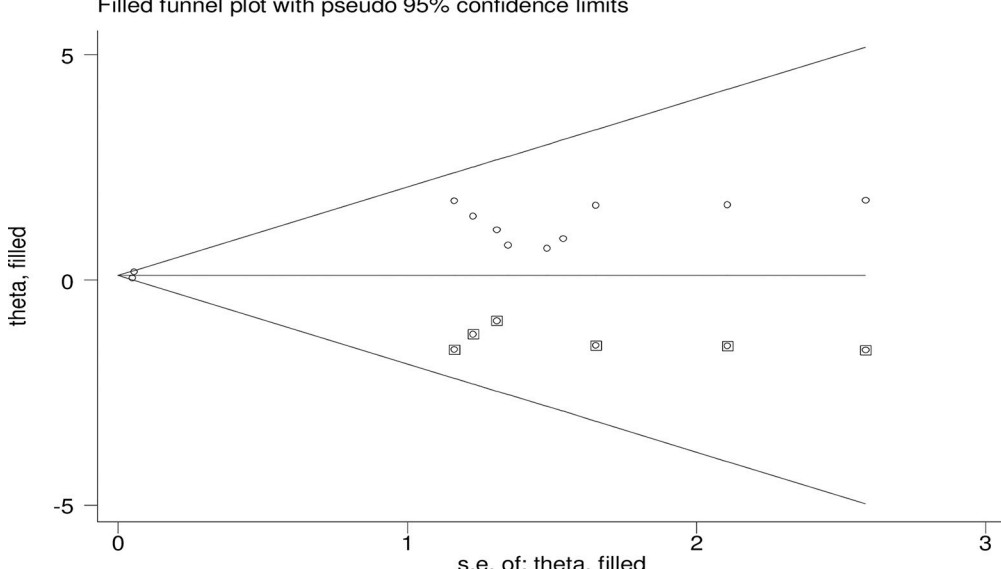

**Fig 4. Trim and fill analysis for the asymmetrical distribution of the included studies.**

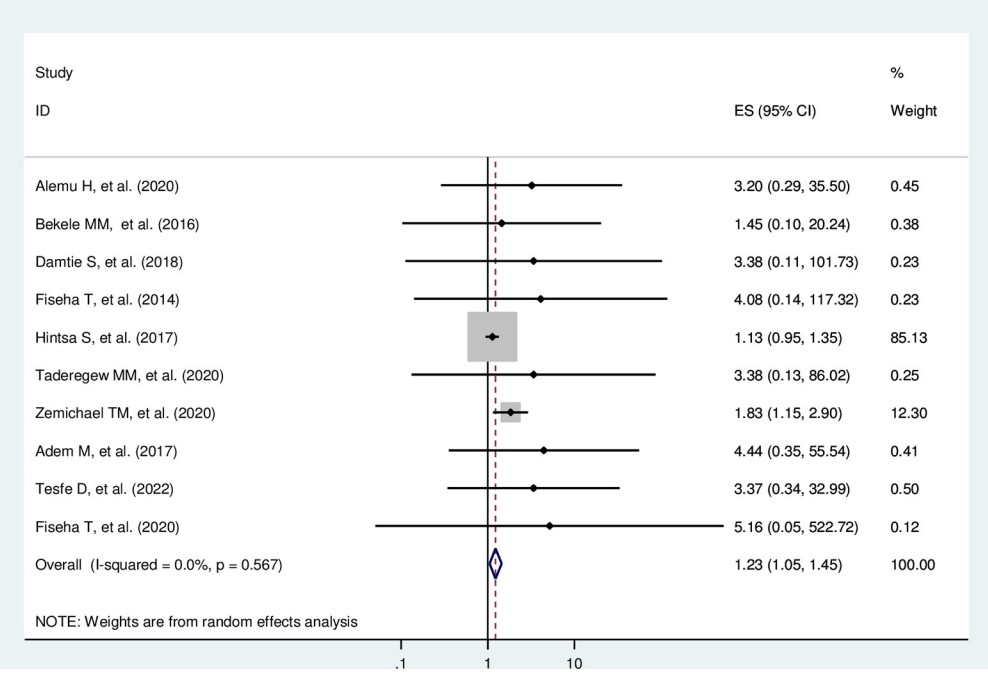

**Fig 5. The pooled effect of duration of diabetes illness on Diabetic nephropathy.**

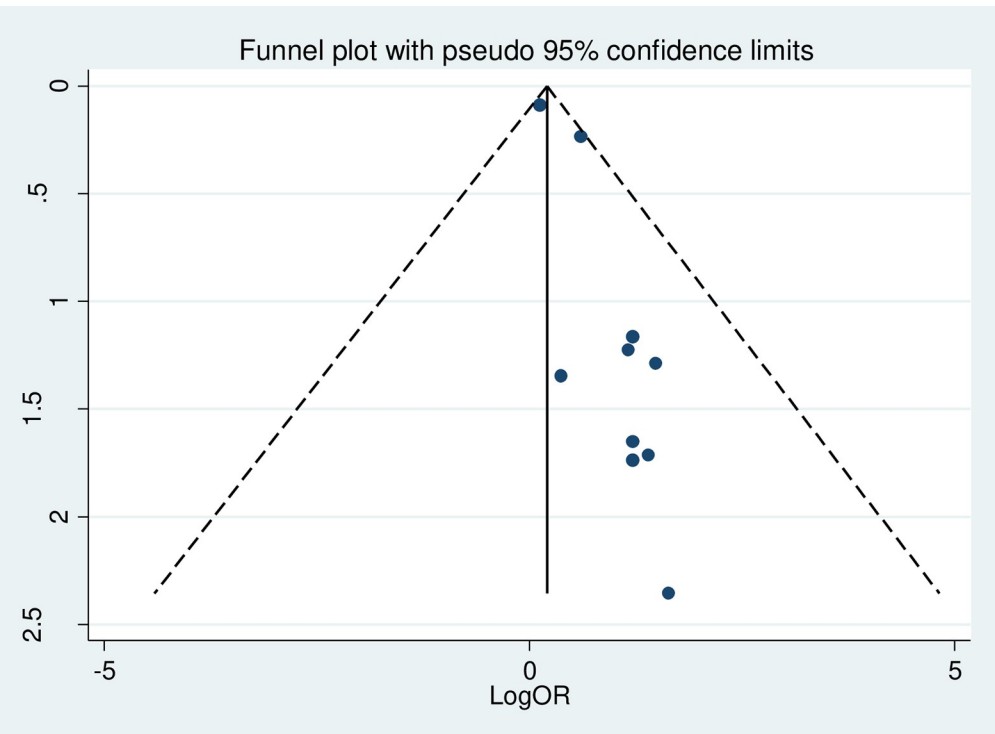

**Fig 6. Funnel plot shows the distribution of the included studies.**

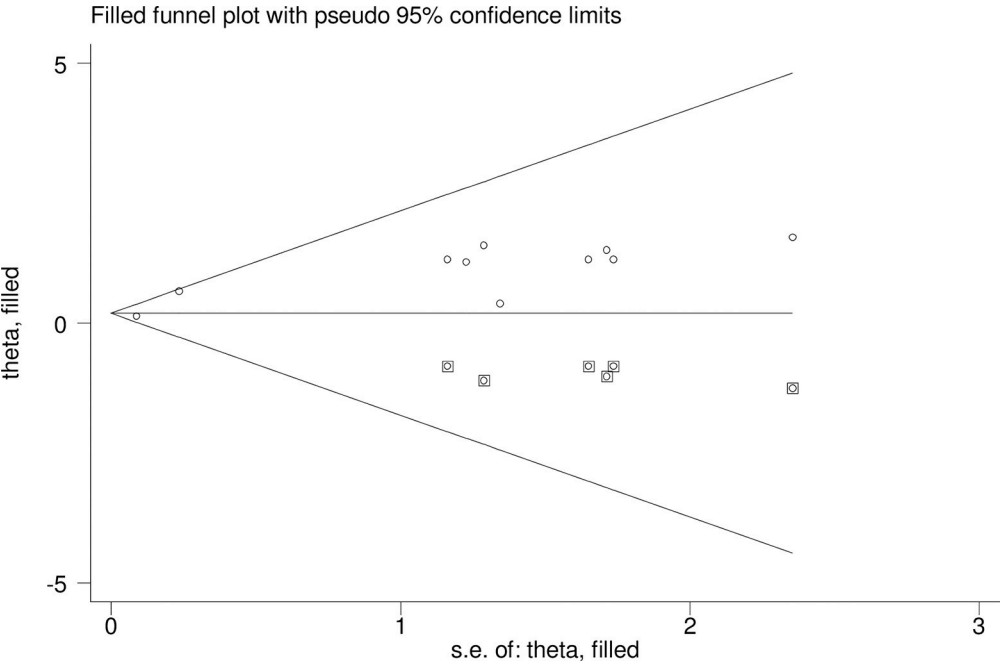

**Fig 7. Trim and fill analysis for the asymmetrical distribution of the included studies.**

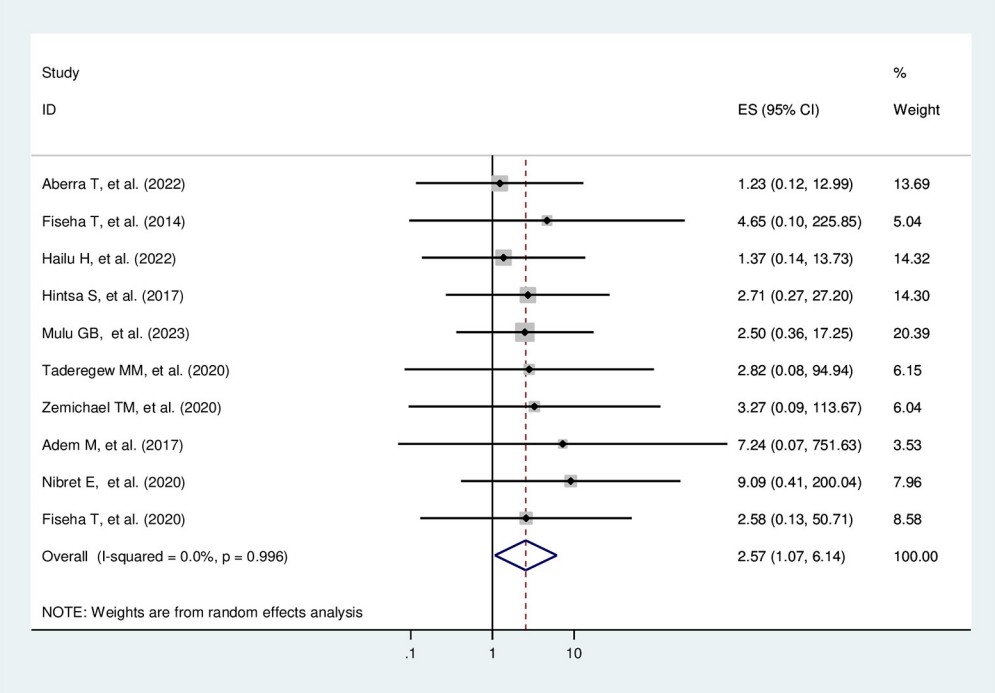

**Fig 8. The pooled effect of poor glycemic control on diabetic nephropathy.**

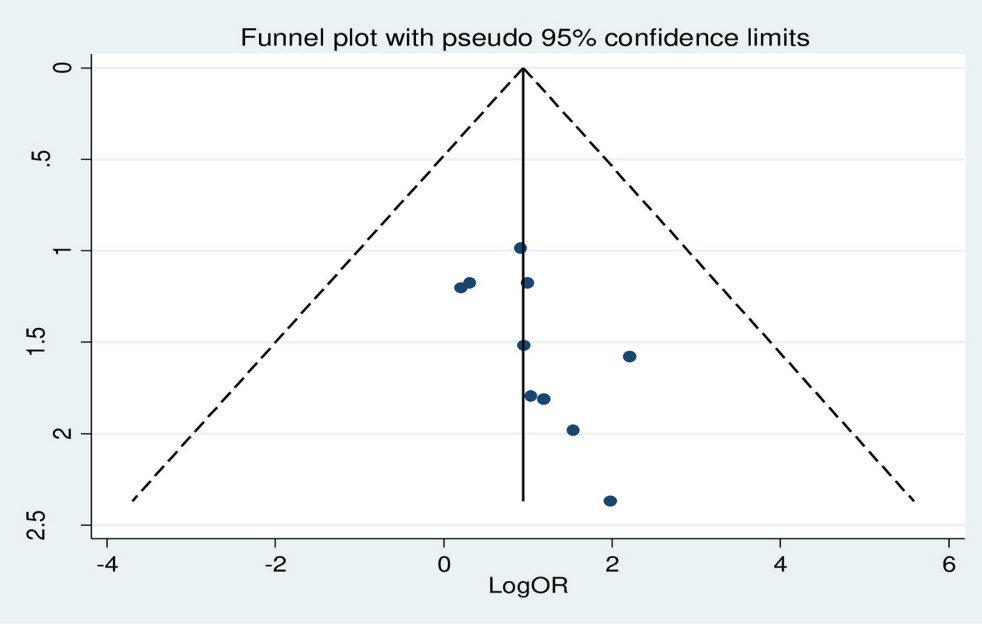

**Fig 9. The funnel plot shows the distribution of the included studies.**

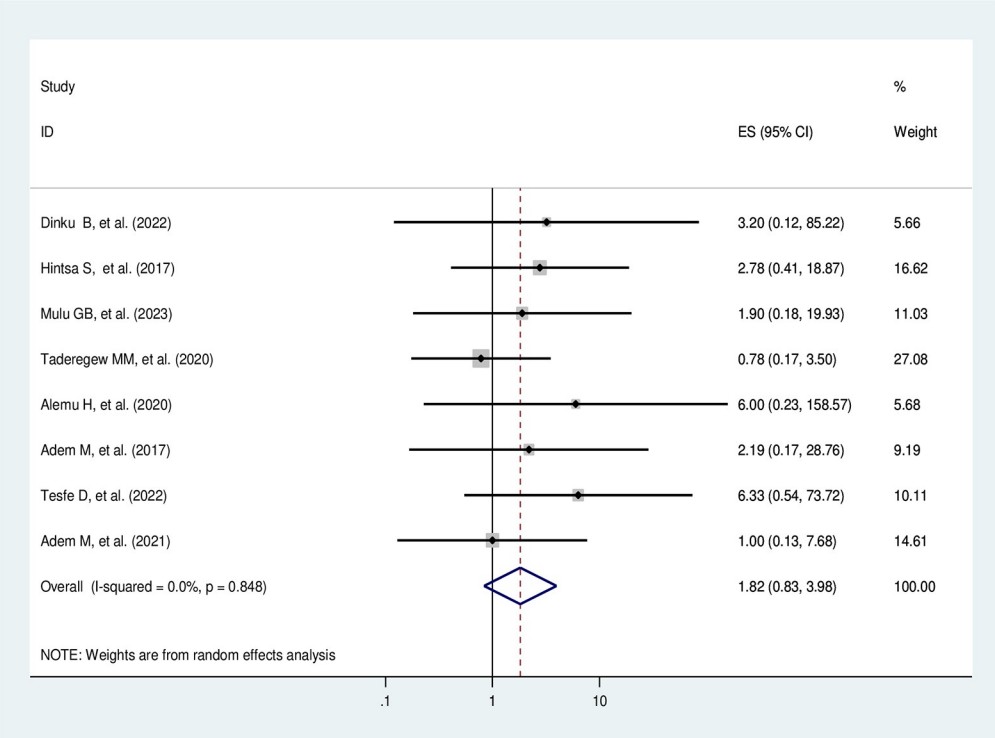

**Fig 10. The pooled effect of elevated systolic hypertension on diabetic nephropathy.**

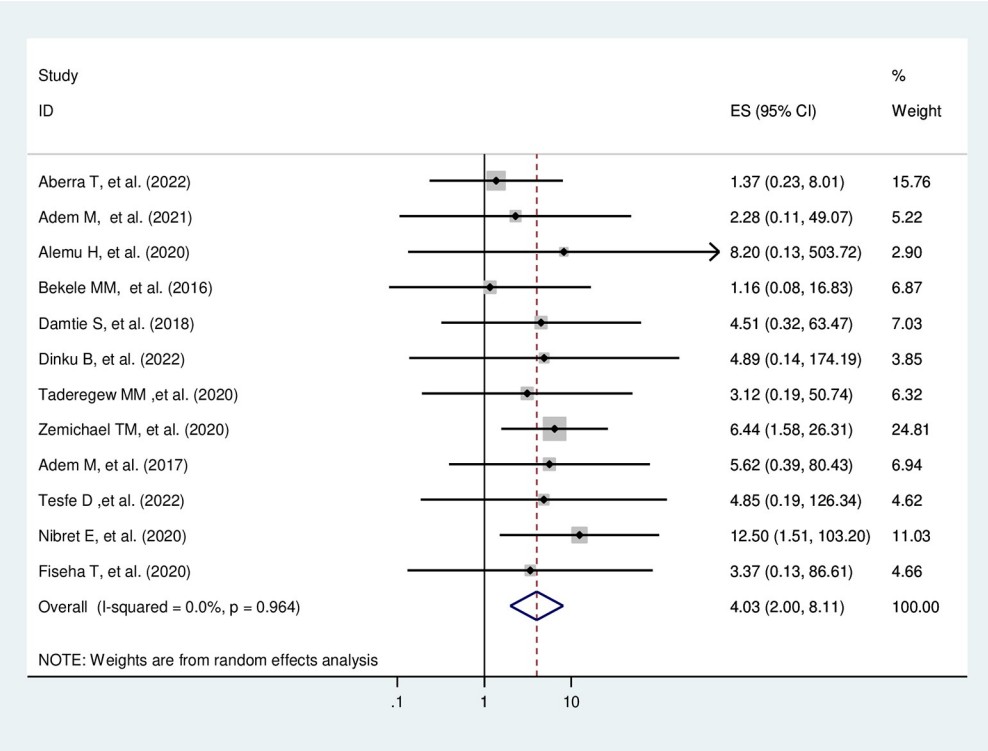

**Fig 11. The pooled effect of co-morbid hypertension on diabetic nephropathy.**

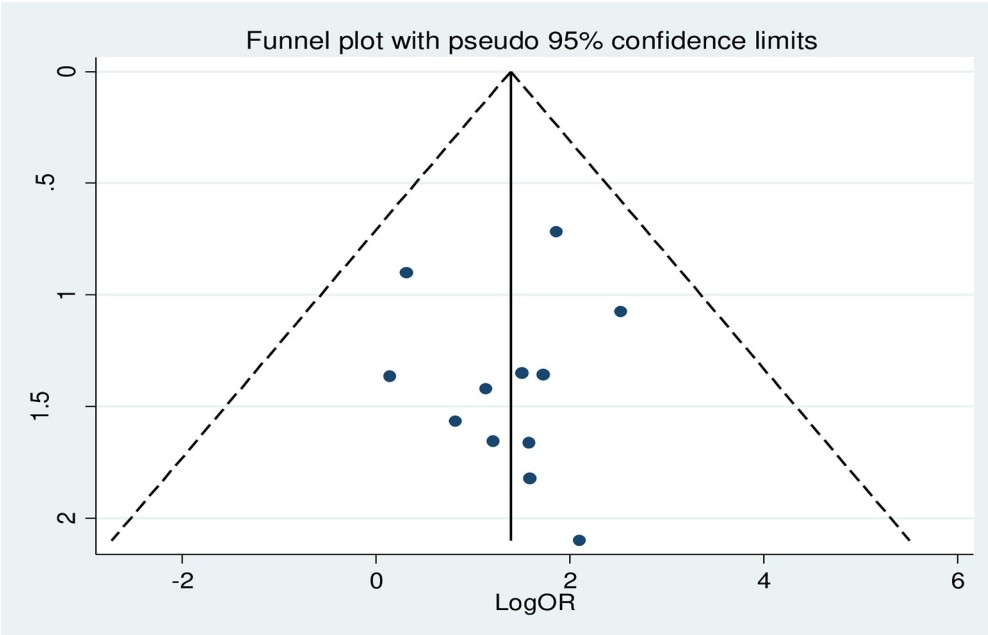

**Fig 12. Funnel plot shows the distribution of the included studies.**

## Conclusions

The findings of this systematic review and meta-analysis study revealed that diabetic patients with advanced age, longer duration of diabetic illness, poor glycemic control status, and having co-morbid hypertension were found to be the determinant factors for DN. Therefore, treatment of co-morbid hypertension and blood glucose and regular screening of renal function should be implemented to reduce the occurrence of DN. Furthermore, healthcare workers should give due attention to diabetic patients with advanced age and a longer duration of diabetic illness to prevent the occurrence of DN.

## Supporting information

**S1 File. PRISMA checklist.**
(DOCX)

**S2 File. Data availability statement.**
(DOCX)

## Acknowledgments

We would like to acknowledge the team members for their invaluable contribution from conception to the final approval for submission to publication.

## Author Contributions

**Conceptualization:** Abere Woretaw Azagew.

**Data curation:** Abere Woretaw Azagew, Zerko Wako Beko, Chilot Kassa Mekonnen.

**Formal analysis:** Abere Woretaw Azagew, Zerko Wako Beko, Chilot Kassa Mekonnen.

**Funding acquisition:** Abere Woretaw Azagew, Zerko Wako Beko.

**Investigation:** Abere Woretaw Azagew, Zerko Wako Beko, Chilot Kassa Mekonnen.

**Methodology:** Abere Woretaw Azagew, Zerko Wako Beko, Chilot Kassa Mekonnen.

**Project administration:** Abere Woretaw Azagew.

**Resources:** Abere Woretaw Azagew.

**Software:** Abere Woretaw Azagew, Zerko Wako Beko, Chilot Kassa Mekonnen.

**Supervision:** Abere Woretaw Azagew.

**Validation:** Abere Woretaw Azagew, Zerko Wako Beko, Chilot Kassa Mekonnen.

**Visualization:** Abere Woretaw Azagew, Zerko Wako Beko, Chilot Kassa Mekonnen.

**Writing – original draft:** Abere Woretaw Azagew, Zerko Wako Beko, Chilot Kassa Mekonnen.

**Writing – review & editing:** Abere Woretaw Azagew, Zerko Wako Beko, Chilot Kassa Mekonnen.

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
