## [Decision Letter · Decision Letter 0]

9 Aug 2023

PONE-D-23-19622Determinants of diabetic nephropathy among diabetic patients in Ethiopia: A systematic review and meta-analysisPLOS ONE

Dear Dr. Azagew,

Thank you for submitting your manuscript to PLOS ONE. After careful consideration, we feel that it has merit but does not fully meet PLOS ONE’s publication criteria as it currently stands. Therefore, we invite you to submit a revised version of the manuscript that addresses the points raised during the review process.

We look forward to receiving your revised manuscript.

Kind regards,

Getu Mosisa

Academic Editor

PLOS ONE

Journal Requirements:

Additional Editor Comments:

 Dear author, please justify the reason why you interested in the title, focus on the objective when you frame your introduction section and you need to revise the discussion section.

Reviewers' comments:

Reviewer's Responses to Questions

**Comments to the Author**

1. Is the manuscript technically sound, and do the data support the conclusions?

Reviewer #1: Yes

Reviewer #2: Partly

2. Has the statistical analysis been performed appropriately and rigorously? 

Reviewer #1: Yes

Reviewer #2: No

3. Have the authors made all data underlying the findings in their manuscript fully available?

Reviewer #1: Yes

Reviewer #2: Yes

4. Is the manuscript presented in an intelligible fashion and written in standard English?

Reviewer #1: Yes

Reviewer #2: No

5. Review Comments to the Author

Reviewer #1: Reviewers comment (Solomon Hintsa) July 22, 2023

1. Your title is timely and uncovered

2. Abstract is advisable to be one page

3. The study population Should be Included in the Methods and in the abstract as well

4. Line 68-70: is the prevalence Global? Regional? or Country level?

5. Line 75: what do you mean to There was no diabetic 76 nephropathy (DN) without being diagnosed with diabetes?

6. Your introduction should focus in Your title. In some of your paragraphs seems that you are Studying on DM rather than DN

7. Line 181: what is the outcome of interest? What is poor quality? Define them

8. Age and duration of diabetes are as continuous in some of the studies and categorized in the other studies. How do you pooled it their respective Odds Ratios?

9. Discussion: it is good to compare with other studies. But, your implications also needed As you did it for duration of illness though it is not detail.

Reviewer #2: the author didn't acknowledged previous work similar with this current submission( https://www.hindawi.com/journals/ijn/2020/8890331/) or the reason for repetition of this study.

6. PLOS authors have the option to publish the peer review history of their article (what does this mean?). If published, this will include your full peer review and any attached files.

Reviewer #1: **Yes: **Solomon Hintsa

Reviewer #2: No

---

## [Author Response · Author response to Decision Letter 0]

13 Oct 2023

The response to the reviewers are attached at the file attachment section.

---

## [Decision Letter · Decision Letter 1]

6 Nov 2023

PONE-D-23-19622R1Determinants of diabetic nephropathy among diabetic patients in Ethiopia: Systematic review and meta-analysisPLOS ONE

Dear Dr. Abere,

Thank you for submitting your manuscript to PLOS ONE. After careful consideration, we feel that it has merit but does not fully meet PLOS ONE’s publication criteria as it currently stands. Therefore, we invite you to submit a revised version of the manuscript that addresses the points raised during the review process.

ACADEMIC EDITOR: Please respond to the comments given by reviewer #2. In addition, abstract section is too shallow please make it comprehensive and informative. 

We look forward to receiving your revised manuscript.

Kind regards,

Getu Mosisa

Academic Editor

PLOS ONE

Journal Requirements:

Reviewers' comments:

Reviewer's Responses to Questions

**Comments to the Author**

1. If the authors have adequately addressed your comments raised in a previous round of review and you feel that this manuscript is now acceptable for publication, you may indicate that here to bypass the “Comments to the Author” section, enter your conflict of interest statement in the “Confidential to Editor” section, and submit your "Accept" recommendation.

Reviewer #1: All comments have been addressed

Reviewer #2: (No Response)

2. Is the manuscript technically sound, and do the data support the conclusions?

Reviewer #1: Yes

Reviewer #2: Yes

3. Has the statistical analysis been performed appropriately and rigorously? 

Reviewer #1: Yes

Reviewer #2: Yes

4. Have the authors made all data underlying the findings in their manuscript fully available?

Reviewer #1: Yes

Reviewer #2: Yes

5. Is the manuscript presented in an intelligible fashion and written in standard English?

Reviewer #1: Yes

Reviewer #2: No

6. Review Comments to the Author

Reviewer #1: This is good title and timley. I have not found ethical problems, dual publications, publication ethics.

Reviewer #2: it is already previously published work, if it has a justification for the repetition of the study, it is worthy if published.

7. PLOS authors have the option to publish the peer review history of their article (what does this mean?). If published, this will include your full peer review and any attached files.

Reviewer #1: **Yes: **Solomon Hintsa

Reviewer #2: No

---

## [Author Response · Author response to Decision Letter 1]

9 Nov 2023

The response to reviewers are attached at the "file attachment" section.

---

## [Editor Report · Decision Letter 2]

29 Nov 2023

PONE-D-23-19622R2Determinants of diabetic nephropathy among diabetic patients in Ethiopia: Systematic review and meta-analysisPLOS ONE

Dear Dr. Azagew,

Thank you for submitting your manuscript to PLOS ONE. After careful consideration, we feel that it has merit but does not fully meet PLOS ONE’s publication criteria as it currently stands. Therefore, we invite you to submit a revised version of the manuscript that addresses the points raised during the review process.

**ACADEMIC EDITOR: Dear author, please respond to the comments provided by reviewer #2 and submit your responses accordingly. **

Please submit your revised manuscript by Jan 13 2024 11:59PM. If you will need more time than this to complete your revisions, please reply to this message or contact the journal office at plosone@plos.org. Please include the following items when submitting your revised manuscript:A rebuttal letter that responds to each point raised by the academic editor and reviewer(s). You should upload this letter as a separate file labeled 'Response to Reviewers'.A marked-up copy of your manuscript that highlights changes made to the original version. You should upload this as a separate file labeled 'Revised Manuscript with Track Changes'.An unmarked version of your revised paper without tracked changes. You should upload this as a separate file labeled 'Manuscript'.If applicable, we recommend that you deposit your laboratory protocols in protocols.io to enhance the reproducibility of your results. Protocols.io assigns your protocol its own identifier (DOI) so that it can be cited independently in the future. For instructions see: https://journals.plos.org/plosone/s/submission-guidelines#loc-laboratory-protocols. Additionally, PLOS ONE offers an option for publishing peer-reviewed Lab Protocol articles, which describe protocols hosted on protocols.io. Read more information on sharing protocols at https://plos.org/protocols?utm_medium=editorial-email&utm_source=authorletters&utm_campaign=protocols.

We look forward to receiving your revised manuscript.

Kind regards,

Getu Mosisa

Academic Editor

PLOS ONE

Journal Requirements:

Additional Editor Comments:

Dear author, please respond to comments provided by reviewer # 2

---

## [Author Response · Author response to Decision Letter 2]

30 Nov 2023

The response to the reviewers are attached in the file attachment section.

---

## [Decision Letter · Decision Letter 3]

27 Dec 2023

Determinants of diabetic nephropathy among diabetic patients in Ethiopia: Systematic review and Meta-analysis

PONE-D-23-19622R3

Dear Mr. Abere,

We’re pleased to inform you that your manuscript has been judged scientifically suitable for publication and will be formally accepted for publication once it meets all outstanding technical requirements.

Kind regards,

Getu Mosisa

Academic Editor

PLOS ONE

Additional Editor Comments (optional):

Reviewers' comments:

Reviewer's Responses to Questions

**Comments to the Author**

1. If the authors have adequately addressed your comments raised in a previous round of review and you feel that this manuscript is now acceptable for publication, you may indicate that here to bypass the “Comments to the Author” section, enter your conflict of interest statement in the “Confidential to Editor” section, and submit your "Accept" recommendation.

Reviewer #2: All comments have been addressed

Reviewer #3: All comments have been addressed

2. Is the manuscript technically sound, and do the data support the conclusions?

Reviewer #2: Yes

Reviewer #3: Yes

3. Has the statistical analysis been performed appropriately and rigorously? 

Reviewer #2: Yes

Reviewer #3: Yes

4. Have the authors made all data underlying the findings in their manuscript fully available?

Reviewer #2: Yes

Reviewer #3: Yes

5. Is the manuscript presented in an intelligible fashion and written in standard English?

Reviewer #2: Yes

Reviewer #3: Yes

6. Review Comments to the Author

Reviewer #2: all comments were addressed by the authors and the paper will be helpful for the scientific community, if published.

Reviewer #3: The way of writing this paper is very good.

Here is my minor comments:

In the abstract- result sub section - Diabetic patients with advanced age (AOR = 1.11, 95% CI: 1.03-120, I2 40 = 0.0%, p = 0.488), - need review this result the correct.

References

Even though, the reference done using software – the list need amendment manually.

7. PLOS authors have the option to publish the peer review history of their article (what does this mean?). If published, this will include your full peer review and any attached files.

Reviewer #2: No

Reviewer #3: **Yes: **Esubalew Tesfahun

---

## [Editor Report · Acceptance letter]

18 Jan 2024

PONE-D-23-19622R3 

PLOS ONE

Dear Dr. Azagew, 

I'm pleased to inform you that your manuscript has been deemed suitable for publication in PLOS ONE. Congratulations! Your manuscript is now being handed over to our production team.

Kind regards, 

on behalf of

Mr Getu Mosisa 

Academic Editor

PLOS ONE